# Macro-Level Modeling of Traffic Crash Fatalities at the Scene: Insights for Road Safety

Carlos Fabricio Assunção da Silva [1,2,*] , Mauricio Oliveira de Andrade [2] , Cintia Campos [3] , Alex Mota dos Santos [4] , Hélio da Silva Queiroz Júnior [2] and Viviane Adriano Falcão [2]

1   Department of Cartographic Engineering, Center of Technologies and Geosciences, Federal University of Pernambuco, UFPE, Avenida Acadêmico Hélio Ramos, Cidade Universitária, s/n, Recife 50740-530, Brazil
2   Department of Civil and Environmental Engineering, Center of Technologies and Geosciences, Federal University of Pernambuco, UFPE, Avenida da Engenharia, s/n-Cidade Universitária, Recife-Pernambuco 50670-420, Brazil; mauricio.andrade@ufpe.br (M.O.d.A.); helio.junior@ufpe.br (H.d.S.Q.J.); viviane.afalcao@ufpe.br (V.A.F.)
3   Faculty of Science and Technology, Federal University of Goiás, Estrada Municipal, Lote 04, Sala 425, Fazenda Santo Antônio, Aparecida de Goiânia 74971-451, Brazil; cintiacampos@ufg.br
4   Center of Agroforestry Sciences and Technologies, Federal University of Southern Bahia, Rodovia Ilhéus/Itabuna, Km 22, Itabuna 45604-811, Brazil; alexmota@ufsb.edu.br
*   Correspondence: carlos.assuncao@ufpe.br

**Abstract:** This study applied 2019 macro-level data from DATASUS to model traffic fatalities at the scene. Ordinary least squares (OLS) and censored regression models (TOBIT) were the methodologies used to identify the significant variables explaining the occurrence of deaths on public roads due to crashes. The number of fatalities on public roadways was then modeled using a multilayer perceptron artificial neural network employing the significant variables as predictors according to the generalization capacity of complex predictive models. The OLS and TOBIT findings indicated that the variables motorcycles and scooters per capita, municipal human development index, and number of SUS emergency units were the most important for modeling traffic fatalities at the scene at the national and regional levels. Applying these variables, the neural network's best results achieved a hit rate of 88% for Brazil and 95% for the Northeast model. The contribution of this study is providing an approach combining various methods and considering a range of variables influencing traffic fatalities at the scene. The findings offer insights for policymakers, researchers, and practitioners involved in road safety initiatives, mainly where crash data are scarce, and macro-level analysis is necessary.

**Keywords:** traffic safety; traffic crash modeling; traffic management; traffic fatalities; multiple linear regression

## 1. Introduction

Traffic crashes are a global problem that can lead to deaths, injuries, and property damage. They are the eighth leading cause of death in the world, with 1.19 million deaths annually [1], lower by 5% compared to the estimated 1.25 million road traffic deaths in 2010, which accounts for nearly 3% of many countries' GDP. Almost 92% of deaths occur in upper-middle, lower-middle, and low-income countries combined, and 56% of the victims are pedestrians, cyclists, or motorcyclists [1].

According to the United Nations [2], many deaths and severe injuries are preventable. Despite improvements in roads and vehicles, in many countries, traffic crashes remain a public health problem that affects society, the environment, and the economy. From this

perspective, the 2023 Agenda [3] included a goal to achieve safer transport systems and reduce the negative impacts of traffic (SDG-11). The UN has also defined 2021 to 2030 as the second Decade of Action for Road Safety, seeking to reduce traffic injuries and deaths worldwide by at least 50%.

According to the WHO [1], road traffic fatalities can be significantly reduced by adopting and enforcing appropriate legislation governing road user behaviors. While many countries have such laws, they often fall short of the WHO's best practices as well as lack consistent implementation and enforcement. Although the safe system approach highlights the importance of system design, behavioral regulations remain crucial in preventing crashes, injuries, and fatalities. Risk factors such as drunk driving, which contributes to 10% of road traffic deaths, as well as the non-use of seat belts, helmets, and child restraint systems, which alone can reduce fatalities by at least 60%, further emphasize the need for stricter enforcement. Additionally, distractions like using communication devices while driving pose significant dangers, reinforcing the critical role of robust and well-enforced traffic laws.

In 2018, Brazil, a continent-sized country, ranked fourth among the countries in the American continent with the most traffic crash deaths, with a rate of 23.4 deaths per 100,000 inhabitants [4]. According to the National Road Safety Observatory [5], one individual passes away every 15 min in the country due to traffic crashes. Between 2009 and 2018, the direct costs of crashes to the Unified Health System (UHS) were about USD 579 million. These numbers reveal the scenario of the impact on health and the economy, which makes road safety a challenge that decisionmakers must consider.

Prediction models are strategically employed in traffic management to forecast the frequency of traffic crashes and aid in decisions on investments in road safety [6]. These traffic crash prediction models can be developed at the micro and macro levels [7]. The micro-level models analyze crashes at segments or intersections, while the macro-level models use data aggregated by area, such as census tract, traffic zone, and municipality [8].

Conventional statistical models, such as Poisson and negative binomial regressions, have been widely used to analyze traffic crashes [9] at the micro [10] and macro levels [11–16]. However, their assumptions lead to inaccurate results in calculating a crash's probability [7,17]. Increasingly, studies are being conducted using machine learning (ML) methods in the transportation field [18] and road safety to identify hotspots [19] and predictors of injury severity [20].

However, despite being less efficient as predictive methods than machine learning methods, statistical regression models identify the relationships between predictor variables and the desired outcome [21–24]. In this way, combining linear or logistic regressions in studies employing machine learning methodologies for predictions or classifications enables the appropriate selection of non-correlated variables that demonstrate statistical significance in generalizing the sought-after responses.

Wang et al. [25] stated that these models could help estimate the frequency of accidents resulting in crashes at a specific location over a given period. Rahman et al. [7] used machine learning (random forest) to show that population density positively affects the number of traffic crash deaths. Delen et al. [20] used a series of artificial neural networks to estimate significant predictors of injury severity, and, as a result, variables had different significance for each injury severity level. Hasan et al. [26] analyzed different ML models to predict crash injury severity in the New Jersey dataset. Random forest and Catboost best predicted fatal, significant, and minor injuries. Vlahogianni et al. [27] suggested using artificial neural networks in a study that compared statistical data and artificial intelligence as the most appropriate method for making forecasts in transportation research.

In addition to identifying crash frequency and severity, these models are also used to identify the factors contributing to frequency or severity. Some of these studies have suggested that demographic factors such as population density, age, and sex as well as the area of the municipality affect the occurrence of traffic crash fatalities [10,28–32]. A study performed by Pljakić et al. [11] showed that increased population density was associated with a higher frequency of deaths. In contrast, another study by Tang et al. [12] showed that municipalities with more paved roads had more crash deaths. Using a decision tree approach, Campos et al. [33] revealed that gross domestic product (GDP) was most closely related to Brazilian traffic death rates. The authors also observed a direct positive relationship between the length of the road network and the vehicle fleet.

In this context, we considered the following research question: Can socioeconomic, infrastructure, traffic, health units, and transport investment variables adequately model, at the municipal level, traffic crash fatalities at the scene? From this perspective, we sought to identify the macro-level characteristics that explain the higher occurrence of deaths at traffic crash sites in certain municipalities or regions, that is, related to the most-severe road crashes. To the best of our knowledge, no exploratory studies have been conducted using machine learning techniques in all Brazilian municipalities yet, specifically for deaths occurring at traffic crash scenes. In this context, this study aimed to develop prediction models for traffic crash fatalities at the scene, grouped by municipality, using various machine learning techniques.

This study contributes to the road safety literature by applying machine learning techniques to analyze a database categorizing traffic fatalities at Brazil's municipal level. It adopted a macro-level data approach to examine incidents occurring at the crash scene. Additionally, it is relevant for managers who work in road safety, public health, and automobile insurance and who seek to predict costs and service capacity as well as perform economic analyses based on crash severity. It also provides insights into the advancement of modeling techniques for predicting fatalities in traffic crashes at the macro level, considering Brazil's continental, cultural, social, and economic dimensions and the lack of an efficient data management policy.

## 2. Materials and Methods

### 2.1. Study Area

Brazil had an estimated population of 212.6 million inhabitants in 2020 [34] in an area of 8,510,345.538 km$^2$ (Figure 1). The country's federal road network accounted for a total length of 75,800 km in 2019, of which 65.4 thousand km corresponded to paved roads [35]. Regarding its vehicular fleet, Brazil had a total of 107,948,371 registered vehicles (cars, light commercials, trucks, buses, motorcycles, scooters, and others) in 2020, according to the National Traffic Secretariat [36].

The country is divided into 26 states [37], and, of these, nine make up the Northeast region. Regarding traffic safety, according to public health records [38], Brazil reported 32,879 traffic deaths, 30.7% of which occurred in the Northeast region states, despite accounting for 26.7% of the population and 12.7% of the vehicle fleet. Table 1 presents information on the population, vehicle fleet, number of municipalities, and traffic fatalities data for each state in Brazil in the year 2019.

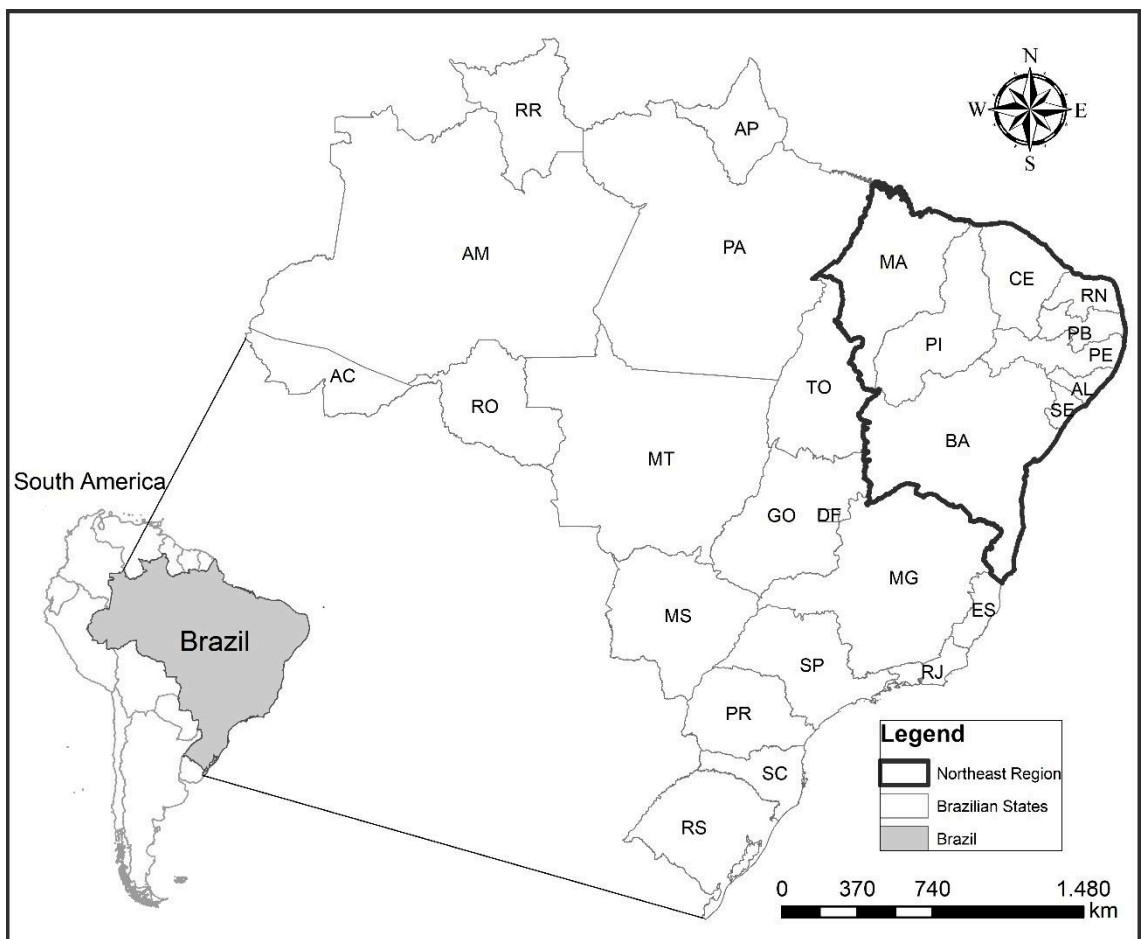

**Figure 1.** Study area.

**Table 1.** Population, vehicle fleet, and traffic fatality data for 2019.

| State | Resident Population | Car Fleet | Deaths on Public Roads | Number of Municipalities |
|---|---|---|---|---|
| Acre (AC) | 881,935 | 91,615 | 119 | 22 |
| Alagoas (AL) * | 3,337,357 | 374,169 | 610 | 102 |
| Amapá (AP) | 845,731 | 85,529 | 85 | 16 |
| Amazonas (AM) | 4,144,597 | 412,140 | 471 | 62 |
| Bahia (BA) * | 14,873,064 | 1,907,497 | 2470 | 417 |
| Ceará (CE) * | 9,132,078 | 1,192,715 | 1640 | 184 |
| Distrito Federal (DF) | 3,015,268 | 1,328,622 | 339 | 1 |
| Espírito Santo (ES) | 4,018,650 | 990,203 | 759 | 78 |
| Goiás (GO) | 7,018,354 | 1,910,006 | 1480 | 246 |
| Maranhão * (MA) | 7,075,181 | 457,104 | 1280 | 217 |
| Mato Grosso (MT) | 3,484,466 | 764,931 | 1038 | 141 |
| Mato Grosso do Sul (MS) | 2,778,986 | 763,091 | 638 | 79 |
| Minas Gerais (MG) | 21,168,791 | 6,467,501 | 3337 | 853 |
| Pará (PA) | 8,602,865 | 631,396 | 1428 | 144 |
| Paraíba (PB) * | 3,996,496 | 552,067 | 774 | 223 |
| Paraná (PR) | 11,433,957 | 4,573,703 | 2433 | 399 |
| Pernambuco (PE) * | 9,557,071 | 1,369,199 | 1511 | 185 |
| Piauí (PI) * | 3,273,227 | 380,035 | 923 | 224 |
| Rio de Janeiro (RJ) | 17,264,943 | 4,646,402 | 1526 | 92 |
| Rio Grande do Norte (RN) * | 3,506,853 | 579,196 | 473 | 167 |
| Rio Grande do Sul (RS) | 11,377,239 | 4,432,248 | 1663 | 497 |

**Table 1.** *Cont.*

| State | Resident Population | Car Fleet | Deaths on Public Roads | Number of Municipalities |
|---|---|---|---|---|
| Rondônia (RO) | 1,777,225 | 298,059 | 382 | 52 |
| Roraima (RR) | 605,761 | 78,387 | 124 | 15 |
| Santa Catarina (SC) | 7,164,788 | 3,053,350 | 1440 | 295 |
| São Paulo (SP) | 44,996,070 | 18,753,362 | 5057 | 645 |
| Sergipe (SE) * | 2,298,696 | 341,946 | 401 | 75 |
| Tocantins (TO) | 1,572,866 | 223,715 | 478 | 139 |

Note: * states in Northeast region, Brazil.

According to De Oliveira et al. [39], despite the Northeast region having the second largest population in the country (Table 1), this region has a municipal human development index (HDI-M) of 0.683, against 0.789 for Brazil and a per capita GDP in 2020 of BRL 19,947, according to IBGE, compared to Brazil's per capita GDP of BRL 35,172.

*2.2. Database*

Brazil does not have a national dataset of traffic crashes, and the official statistics from national road safety include the number of deaths. Thus, the data applied in this study were from public sources for the 5570 municipalities in Brazil (1794 in the Northeast), organized into 26 states and the Federal District (Figure 1). The data were from the Brazilian Institute of Geography and Statistics (IBGE), the Ministry of Health, and the Central Bank of Brazil and applied in the study as collected initially or treated as indices, as presented in Table 2.

**Table 2.** Description of the response and explanatory variables obtained for each municipality.

| Variable Group | Variable Name | Description | Year | Unit | Variable Group |
|---|---|---|---|---|---|
| Response Variable | Deaths on public roads | Deaths on public roads due to traffic crashes | 2019 | Absolute number | [38] |
| Explanatory Variables | Road fatality rate | Deaths divided by population multiplied by 100,000 | 2019 | Deaths/100,000 inhabitants | [38,40] |
| | Deaths by type | Deaths resulting from a specific incident | 2019 | Absolute number | [40] |
| | Deaths by Occurrence rate | Deaths by occurrence divided by the population of a given area, then multiplied by 100,000 inhabitants. | 2019 | Deaths/100,000 inhabitants | [38,40] |
| | Cars per capita | Motorized vehicles per inhabitant | | Vehicles/inhabitant | [36,40,41] |
| | Motorcycles and scooters per capita | Motorcycles and scooters per inhabitant | 2019 | Motorcycle/inhabitant | [36,40,41] |
| | Road extension by cars | Total road mileage divided by the number of cars | 2019 | km/car | [36,42] |
| | Road extension by motorcycles | Total road mileage divided by the number of motorcycles | 2019 | km/motorcycle | [36,42] |
| | Municipal human development index | A composite measure of indicators from three dimensions of human development: longevity, education, and income | 2019 | Index | [41] |

**Table 2.** *Cont.*

| Variable Group | Variable Name | Description | Year | Unit | Variable Group |
|---|---|---|---|---|---|
| Explanatory Variables | Road extension by municipality | Length of roads per municipality road mileage in km | 2019 | km | [40,42] |
| | Road density | Road Mileage in km per inhabitant | 2020 | km/inhabitant | [40,42] |
| | Investment in road infrastructure per capita | Monetary values (in Brazilian reals) invested in road infrastructure per inhabitant | 2020 | R$/inhabitant | [43] |
| | Investment in housing and urban development per capita | Monetary values (in real) invested in housing and urbanization per inhabitant | 2019 | R$/inhabitant | [43] |
| | GDP per capita | Gross domestic product per capita | 2019 | R$/inhabitant | [40,43] |
| | Demographic density | Number of people divided by the area of the municipality | 2019 | Inhabitants/km$^2$ | [40] |
| | SUS emergency units | Number of health units with emergency care | 2019 | Absolute number | [38] |

Length of roads per municipality. Mileage in km.

The response variable used in this study corresponded to the number of road deaths due to traffic crashes in Brazilian municipalities in 2019. These data were obtained from the Mortality Information System of the Unified Health System (DATASUS) and originated from the "Death Declaration" document. The indices of traffic deaths per inhabitant and vehicle allowed us to know the negative impact of traffic crashes and measure traffic safety [44]. In this way, modeling the number of deaths made it possible to identify the macro-level factors related to more severe crashes, identifying the socioeconomic and road infrastructure characteristics of the most-exposed municipalities.

Considering that traffic crashes are rare and random [9], a set of variables was selected, according to data availability, to assess the relationship with the occurrence of the most severe traffic crashes, that is, with deaths at the site. Thus, the explanatory variables corresponded to the socioeconomic characteristics of the municipality of occurrence, vehicle fleet composition, and road infrastructure. Table 3 presents the descriptive statistics of the dataset.

**Table 3.** Descriptive data statistics.

| Variable | Mean | Median | Stan. Dev. | Min | Max |
|---|---|---|---|---|---|
| Deaths on public roads | 3.00 | 1.00 | 6.85 | 0.00 | 242.00 |
| Road fatality rate | 13.71 | 8.18 | 22.26 | 0.00 | 585.50 |
| Deaths by occurrence | 5.91 | 2.00 | 21.58 | 0.00 | 763.00 |
| Deaths by occurrence rate | 18.40 | 13.09 | 30.46 | 0.00 | 1099.00 |
| Cars per capita | 0.20 | 0.18 | 0.21 | 0.00 | 11.56 |
| Motorcycles and scooters per capita | 0.14 | 0.13 | 0.10 | 0.00 | 5.61 |
| Extension of highways by cars | 0.66 | 0.24 | 2.36 | 0.00 | 105.90 |
| Extension of highways by motorcycles | 0.45 | 0.29 | 0.88 | 0.01 | 41.24 |
| HDI-M | 0.66 | 0.67 | 0.08 | 0.00 | 0.86 |

**Table 3.** *Cont.*

| Variable | Mean | Median | Stan. Dev. | Min | Max |
|---|---|---|---|---|---|
| Road extension by municipality | 660.70 | 415.60 | 874.90 | 16.27 | 20,918.00 |
| Road density | 1.35 | 1.02 | 1.60 | 0.00 | 20.93 |
| Investment in transportation per capita | 124.60 | 33.10 | 241.60 | 0.00 | 4498.00 |
| Investment in housing and urban development per capita | 300.50 | 243.90 | 262.30 | 0.00 | 3645.00 |
| GDP per capita | 24.55 | 18.19 | 25.55 | 4.48 | 464.90 |
| Demographic density | 120.00 | 25.07 | 627.40 | 0.05 | 14,208.00 |
| SUS emergency units | 1.87 | 1.00 | 6.18 | 0.00 | 303.00 |

*2.3. Methodological Steps*

This study aggregated municipal data and spatial information to examine how socioeconomic, fleet, and infrastructure factors related to traffic crash fatalities on Brazilian roads. It used two analytical approaches. The first encompassed all Brazilian municipalities and the second only the municipalities in the Northeastern region. This distinction aimed to verify the proposed models' ability to predict road crash deaths in different contexts since Brazil is a continent-sized country. The diversity of the Brazilian regions in terms of economic aspects and institutional capacity could lead to different results in terms of the significance of the variables.

The flowchart presented in Figure 2 demonstrates the methodological steps applied in this study. It involved a data survey, preliminary treatment, and database formation, as well as the establishment of the explanatory variables to be submitted to the prediction and classification models by machine learning and artificial neural networks (ANNs). Thus, the flowchart was followed for all municipalities and then only for the municipalities in the Northeast region, as detailed in the following sections.

2.3.1. Database Treatment

The database, composed of the variables described in Table 1, was subjected to a data treatment step of the original variables. Thus, a conventional machine learning pipeline, consisting of data mining, data cleaning, pre-processing, removal of discrepant values, resource selection, model training, and validation, was used to predict the number of fatal failures.

First, we standardized the data to ensure comparability across variables with different units. This process involved transforming each variable with a mean of zero and a standard deviation of one. Standardization normalizes the scale of the variables, facilitating more accurate analysis and comparison. Subsequently, we applied the isolation forest algorithm (iForest) to identify and eliminate outliers in the dataset, and we generated a linear regression (LR) using all the variables to calculate the mean absolute error (MAE) of the model and serve as the baseline. The isolation forest (iForest) algorithm detects anomalies by constructing a random forest of decision trees [45]. Tailored for isolating anomalies within a dataset, it exploits the characteristic that anomalies often require fewer connections, allowing efficient identification through iterative partitioning [46]. This unique approach ensures fast computation and robust performance across diverse domains, making iForest a tool for anomaly detection tasks [45]. Outliers can skew statistical measures, leading to poor fit and lower predictive ability, which takes advantage of isolation to exploit subsampling with low memory requirements. The dataset was divided into training and testing subsets to ensure the model was validated based on unknown data. This separation helped prevent data leakage by avoiding any influence of the test data on the training process.

We used the iForest algorithm with a default contamination rate of 10% to identify outliers in the dataset. We removed the outliers and then randomly split the remaining data into training and test sets in a 70%:30% ratio.

To reduce model complexity and possible existing correlations and to select the most critical variables in the dataset, we used a subset of input features most relevant to the target response—the number of road fatalities in a municipality, using two different approaches: correlation statistics and mutual information statistics.

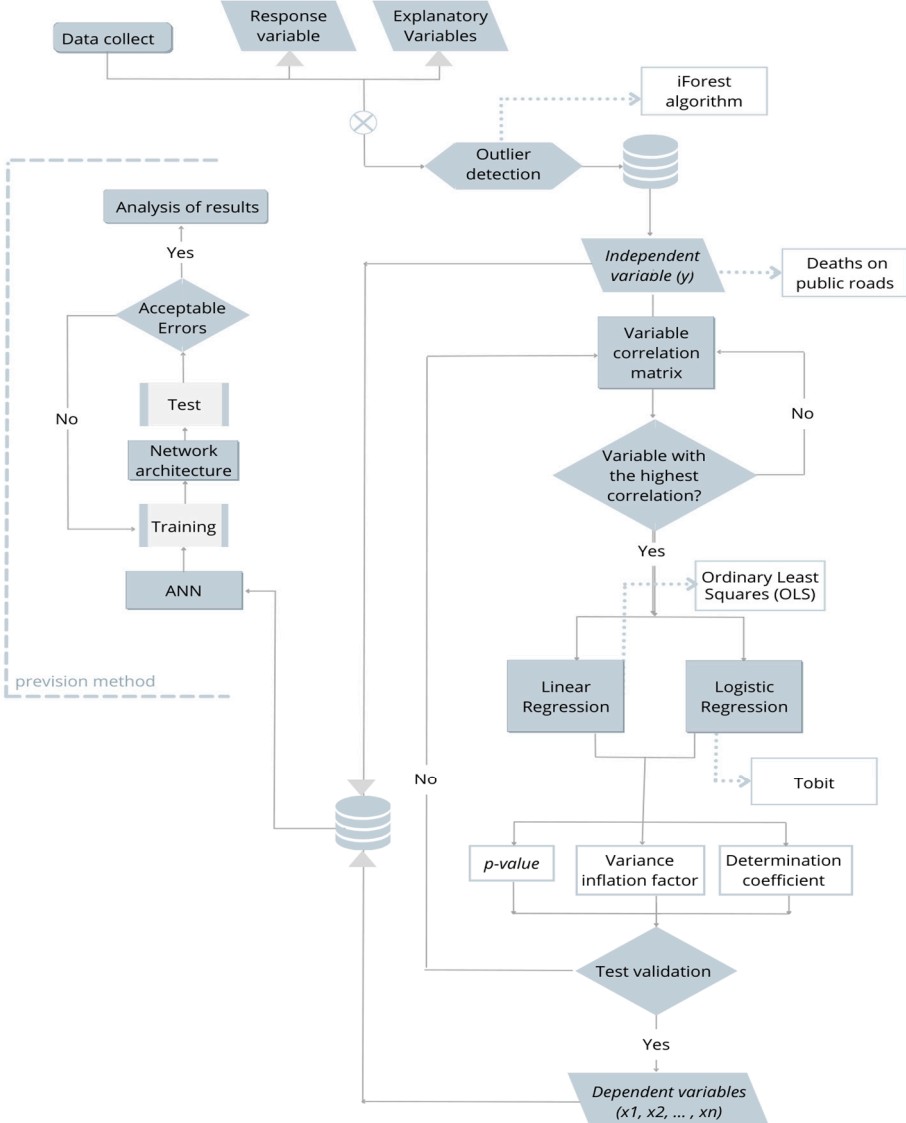

**Figure 2.** Flowchart of the methodological steps applied in this research.

### 2.3.2. Correlation Statistics and Linearity Verifications

We used Pearson's correlation matrices to identify the variables most correlated with this study's response variable. According to Evans [47], we discarded any variable with a correlation of less than 0.20 because it was negligible. Additionally, we performed graphical analyses using scatter plots with trend lines to evaluate the strength and direction of the linear relationships among the variables.

### 2.3.3. Regression Models and Statistical Validation

We applied two regression methods to the database (ordinary least squares (OLS) and TOBIT regression) to select the explanatory variables related to the evolution of the cases of deaths on public roads and their subsequent application in the neural network models.

We validated the OLS regression model using hypothesis tests for the model conditions. Thus, we tested the heteroscedasticity of the data through White's test to find the similarities between the errors found for the system's selected inputs. Another test used was the variance inflation factor (VIF), which tests collinearity among variables. The last test required for OLS validation was the residual normality test, where histograms checked some trends among the residuals.

The truncated dependent variable regression model, or TOBIT, has as its principle a correlation in which the dependent variable concentrate between ranges of values, that is, points equal to a threshold value [48]. Like OLS, the TOBIT model requires hypothesis tests that validate the model and the coefficients raised for the variables, these being the VIF and the normality of the residuals, just like the previous model.

The TOBIT regression model, also known as a truncated or censored regression model, is used when the dependent variable is truncated at some point, typically at zero [49]. The dependent variable's distribution is incomplete because some values are censored or unobserved below a certain threshold. The TOBIT model allows for the estimation of regression parameters by adjusting to this specific data structure. In this study, we applied TOBIT to address the truncated nature of the dependent variable: the number of deaths on public roads. This variable was truncated because it could not take on negative values and, in many cases, could be equal to zero (when there were no deaths in a particular location or period). The TOBIT model is suitable for handling this type of data as it accounts for the censoring in the distribution of the dependent variable.

### 2.3.4. Predicting Occurrences with Artificial Neural Networks (ANNs)

A supervised artificial neural network method, multi-layer perceptron (MLP), was employed to examine the influence of the variables selected in the preceding steps on the occurrence of deaths on public roads due to traffic crashes from the analysis of the prediction of occurrences.

Artificial neural networks generalize predictions to responses still unknown to the network. Schutz et al. [50] highlighted this network model's recurrent approach for discrete time-series value prediction.

Luger [51] suggested that, through the construction of concepts and representative analysis of a predicted event, coupled with the data and objectives to be portrayed by the model, the machine learning system recognizes behavior patterns, referred to as the potential for generalization. It is also capable of attributing the acquired knowledge to similar cases.

Specifically, the supervised neural networks of the multi-layer perceptron (MLP) type used backpropagation as its base algorithm, and we validated the network by verifying the coefficient of determination of the training and test stages.

We tested the net architectures to define the configuration with the highest prediction accuracy. We also defined this analysis using the following information: the number of intermediate layers in the system, the number of neurons per hidden layer, the activation function between layers of neurons, and the mathematical model for validating the results.

Even though there is no consensus in the literature regarding the ideal percentage distribution of data for training and testing in analyses with artificial neural networks (ANNs), several approaches have been adopted. For instance, Chen et al. [52] chose a distribution of 80% for training and 20% for testing, while other studies opted for a ratio of

60% for training and 40% for testing [53]. These references influenced the decision on the distribution in this study, resulting in an allocation of 70% for training and 30% for testing.

The Levenberg–Marquardt backpropagation feeding algorithm, the moment-weighted gradient regression adaptation function, and the bias learning function were used, and their validation was defined by the mean squared error (MSE).

The artificial neural network (ANN) model was developed after the linear regression model because linear models, like ordinary least squares (OLS), effectively identify significant linear relationships between predictor and response variables. These models provide a solid foundation for variable selection by pinpointing which variables have statistically significant relationships with the variable of interest. Following this selection process, the ANN model was employed to capture complex, non-linear patterns in the data, which are beyond the reach of simple linear models. This approach allows the ANN to model intricate relationships between input and output variables that linear models cannot adequately address.

## 3. Results

After scaling the variables, a linear regression was trained as the baseline model with a mean absolute error of 36.640 for the test set. Next, the iForest algorithm detected and removed 373 municipalities as outliers. After removing the outliers, a new linear regression model was trained, leading to an MAE of 34.546. This suggested that excluding outliers from the dataset led to a lower MAE. This result demonstrated that the model without outliers showed a predicted value closer to the true one.

After excluding the outlier municipalities, the Pearson correlation coefficient (R) was calculated to identify how the variables in this study correlated with the number of deaths on public roads. Thus, the coefficients are presented for the two datasets: Brazil (Table 4) and Northeastern Brazil (Table 4). Table 5 presents the identification of the variables corresponding to the codes present in the matrices in Figure 3.

**Table 4.** Pearson correlation coefficient matrices for the Brazil dataset (a) and the Northeast region (b).

| | | | | | | | | **(a)** | | | | | | | | |
|---|---|---|---|---|---|---|---|---|---|---|---|---|---|---|---|---|
| **Variables** | **1** | **2** | **3** | **4** | **5** | **6** | **7** | **8** | **9** | **10** | **11** | **12** | **13** | **14** | **15** | **16** |
| **1** | 1 | | | | | | | | | | | | | | | |
| **2** | 0.1 | 1 | | | | | | | | | | | | | | |
| **3** | 0.9 | 0 | 1 | | | | | | | | | | | | | |
| **4** | 0.2 | 0.9 | 0.1 | 1 | | | | | | | | | | | | |
| **5** | 0.1 | 0 | 0.1 | 0 | 1 | | | | | | | | | | | |
| **6** | 0.1 | 0 | 0 | 0 | 0.6 | 1 | | | | | | | | | | |
| **7** | −0.1 | 0 | 0 | 0 | −0.2 | −0.1 | 1 | | | | | | | | | |
| **8** | −0.1 | 0.1 | −0.1 | 0.1 | −0.1 | −0.2 | 0.6 | 1 | | | | | | | | |
| **9** | 0.2 | 0 | 0.2 | 0.1 | 0.6 | 0.2 | −0.3 | −0.1 | 1 | | | | | | | |
| **10** | 0.7 | 0 | 0.7 | 0.1 | 0.1 | 0.1 | 0 | 0 | 0.2 | 1 | | | | | | |
| **11** | 0.3 | 0.1 | 0.4 | −0.1 | 0.2 | 0 | −0.1 | −0.1 | 0.3 | 0.2 | 1 | | | | | |
| **12** | −0.1 | 0.1 | −0.1 | 0 | 0.3 | 0.1 | 0 | 0.1 | 0.3 | −0.1 | 0 | 1 | | | | |
| **13** | 0 | 0.1 | 0 | 0 | 0.2 | 0 | 0 | 0 | 0.3 | 0 | 0.1 | 0.1 | 1 | | | |
| **14** | 0.1 | 0.1 | 0.1 | 0.1 | 0.3 | 0 | −0.1 | 0 | 0.4 | 0.2 | 0.2 | 0.2 | 0.4 | 1 | | |
| **15** | 0.3 | −0.1 | 0.4 | 0 | 0.1 | 0 | 0 | −0.1 | 0.2 | 0.2 | 0.8 | −0.1 | 0 | 0.1 | 1 | |
| **16** | 0.8 | 0 | 0.9 | 0 | 0.1 | 0 | 0 | −0.1 | 0.2 | 0.6 | 0.4 | −0.1 | 0 | 0.1 | 0.4 | 1 |

**Table 4.** *Cont.*

| Variables | 1 | 2 | 3 | 4 | 5 | 6 | 7 | 8 | 9 | 10 | 11 | 12 | 13 | 14 | 15 | 16 |
|---|---|---|---|---|---|---|---|---|---|---|---|---|---|---|---|---|
| | | | | | | | **(b)** | | | | | | | | | |
| 1 | 1 | | | | | | | | | | | | | | | |
| 2 | 0.3 | 1 | | | | | | | | | | | | | | |
| 3 | 0.8 | 0 | 1 | | | | | | | | | | | | | |
| 4 | 0.3 | 0.9 | 0.2 | 1 | | | | | | | | | | | | |
| 5 | 0.4 | 0 | 0.4 | 0.1 | 1 | | | | | | | | | | | |
| 6 | 0.2 | 0.1 | 0.1 | 0.2 | 0.5 | 1 | | | | | | | | | | |
| 7 | −0.1 | 0 | −0.1 | 0 | −0.3 | −0.2 | 1 | | | | | | | | | |
| 8 | −0.2 | 0 | −0.1 | 0 | −0.2 | −0.4 | 0.7 | 1 | | | | | | | | |
| 9 | 0.5 | 0 | 0.4 | 0.1 | 0.6 | 0.3 | −0.2 | −0.3 | 1 | | | | | | | |
| 10 | 0.6 | 0 | 0.5 | 0.2 | 0.3 | 0.1 | 0.1 | 0.2 | −0.3 | 1 | | | | | | |
| 11 | 0.4 | −0.1 | 0.5 | −0.1 | 0.5 | 0 | −0.2 | −0.1 | 0.2 | 0.2 | 1 | | | | | |
| 12 | 0 | 0 | 0 | 0 | −0.1 | 0 | 0.1 | 0.1 | −0.1 | 0 | −0.1 | 1 | | | | |
| 13 | 0 | 0 | 0 | 0 | 0.2 | 0 | 0 | 0.1 | 0.1 | 0 | 0.1 | 0.1 | 1 | | | |
| 14 | 0.2 | 0 | 0.1 | 0.1 | 0.2 | 0.1 | 0 | 0.1 | 0.3 | 0.3 | 0.2 | 0.2 | 0.3 | 1 | | |
| 15 | 0.4 | −0.1 | 0.6 | 0 | 0.3 | 0 | −0.1 | −0.1 | 0.4 | 0.2 | 0.8 | 0 | 0 | 0.1 | 1 | |
| 16 | 0.7 | 0 | 0.8 | 0.1 | 0.4 | 0.1 | −0.1 | −0.1 | 0.4 | 0.5 | 0.5 | 0 | 0 | 0.2 | 0.7 | 1 |

**Table 5.** Variables used in the correlation matrices and their respective codes.

| Variable | Code |
|---|---|
| Deaths on public roads | 1 |
| Road fatality rate | 2 |
| Deaths by occurrence | 3 |
| Deaths by occurrence rate | 4 |
| Cars per capita | 5 |
| Motorcycles and scooters per capita | 6 |
| Extension of highways by cars | 7 |
| Extension of highways by motorcycles | 8 |
| HDI-M | 9 |
| Road extension by municipality | 10 |
| Road density | 11 |
| Investment in transportation per capita | 12 |
| Investment in housing and urban development per capita | 13 |
| GDP per capita | 14 |
| Demographic density | 15 |
| SUS emergency units | 16 |

The dependent variable, deaths on public roads (1), obtained the highest Pearson correlation coefficients with identical explanatory variables on both datasets, (Table 4a) Brazil and (Table 4b) Northeast Region.

The correlation matrix for Brazil (Table 4a) showed that the dependent variable (1) had strong positive correlations with deaths by occurrence (3), road extension per municipality (10), and SUS emergency units (16). The first two could be correlated as they were related to exposure. The emergency units, on the other hand, could be related to the population of the municipalities since it also showed a high and positive correlation with the variable's road density (11) and demographic density (15). Among the other variables, we observed a positive correlation between the municipal human development index (HDI-M) (9), with cars per capita (5), GDP per capita (14), investment in housing, and urbanism per capita (13), road density (11), and investment in transportation per capita (12), indicating a possible socioeconomic relationship of the municipalities.

a

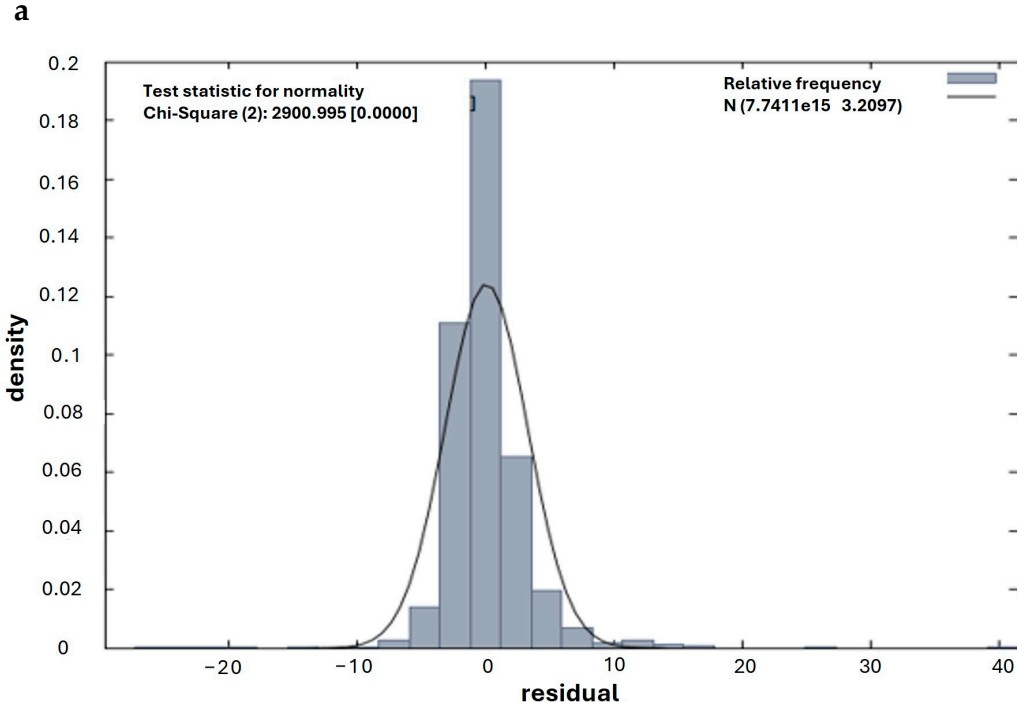

b

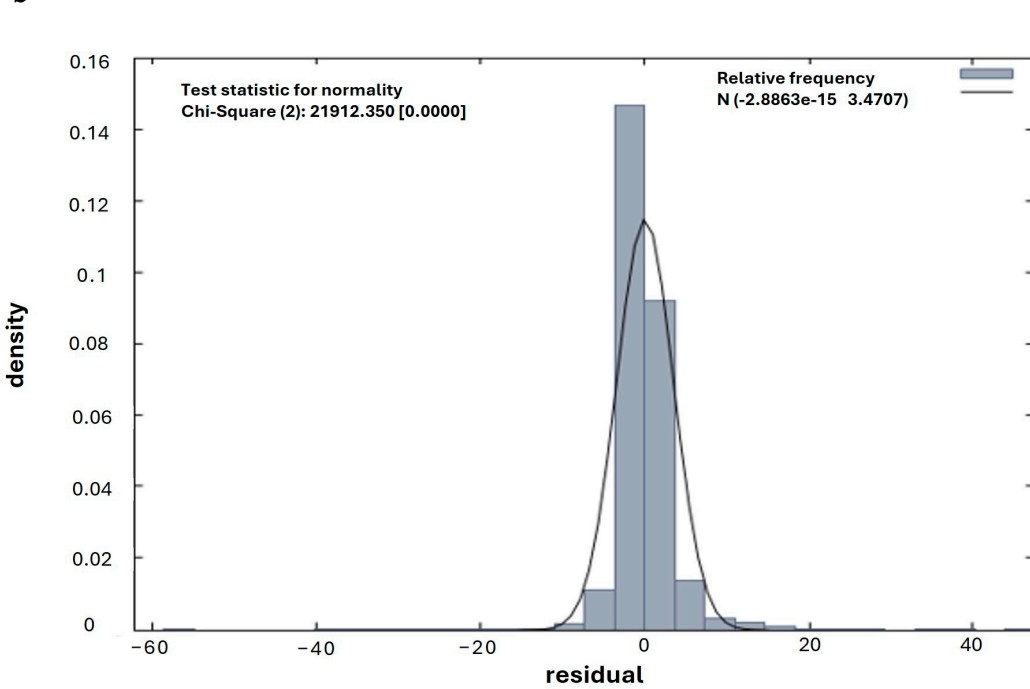

**Figure 3.** Histogram of distribution of residuals, normality test of residuals for data density: (**a**) Brazil and (**b**) Northeast.

For the Northeast region of Brazil (Tabel 4b) the HDI-M (9) presented a stronger correlation with the dependent variable (1) when compared to the national scenario (Table 4a). The other identified correlations were similar to the national scenario; however, there as a higher negative correlation between motorcycles and scooters per capita (6) and the road extension by motorcycles (8) in the Northeast region. This result could be related to the high number of motorcycles registered in the Northeast region. Furthermore, contrary to the national reality, investments in transportation per capita (12) had a negative relationship with the HDI-M (9).

Thus, to select parameters with the most significant influence on the occurrence of deaths on public roads, a new evaluation of the variables' relationships was performed using other techniques to corroborate the present analysis.

### 3.1. Mutual Information Statistics

The OLS and TOBIT regression models were applied to define the parameters that better described the dependent variable. After analyzing the metrics of the models, Table 6 presents the ones that obtained the best results.

**Table 6.** Results of the OLS and TOBIT models for Brazil and Northeast region data.

| Data Set Models | Brazil | | | | | Northeast Region | | | | |
| | OLS | | TOBIT | | | OLS | | TOBIT | | |
| Variable | Coef | *p*-Value | Coef | *p*-Value | VIF | Coef | *p*-Value | Coef | *p*-Value | VIF |
|---|---|---|---|---|---|---|---|---|---|---|
| Constant | −2.907 | <0.001 | −6.14512 | <0.001 | - | −6.682 | <0.001 | −8.942 | <0.001 | - |
| Motorcycles and scooters per capita | 1.127 | 0.0144 | 2.242 | <0.001 | 1.049 | 7.344 | 0.0021 | 10.025 | 0.0011 | 1.495 |
| HDI-M | 4.332 | <0.001 | 7.378 | <0.001 | 1.241 | 3.323 | 0.0049 | 4.105 | 0.0062 | 1.711 |
| Road extension per municipality | 0.003 | <0.001 | 0.0036 | <0.001 | 1.702 | 0.002 | <0.001 | 13.452 | <0.001 | 1.373 |
| Investment in transportation per capita | −0.001 | <0.001 | −0.0039 | <0.001 | 1.115 | −0.0002 | <0.001 | −0.003 | <0.001 | 1.055 |
| Investment in housing and urban development per capita | −0.0004 | 0.0217 | −0.0012 | <0.001 | 1.077 | −0.002 | 0.090 | −0.004 | <0.001 | 1.088 |
| Demographic density | 0.0002 | 0.0767 | 0.0003 | 0.0162 | 1.275 | $-6.75 \times 10^{-6}$ | <0.001 | 0.002 | 0.0369 | 1.095 |
| SUS emergency health units | 0.607 | <0.001 | 0.583 | <0.001 | 1.940 | 0.054 | <0.001 | 0.542 | <0.001 | 2.321 |

* Note: for the same set of variables, the results observed for the VIF metric for the Brazil and Northeast models did not vary according to the regression established model.

The regression coefficients of the models for the national and regional analysis, Table 6, presented positive correlations between the occurrence of deaths on the roads and the variable motorcycles and scooters per capita, the local socioeconomic variable, and the HDI-M. We highlight, however, the higher coefficient for the motorcycle and scooter per capita variable for the Northeast region, showing that this type of vehicle represented an important explanatory factor for the severity of crashes in this region compared to the result achieved in the model for Brazil.

Investments in housing, urban infrastructure, and transportation had an opposite relationship with the incidence of traffic crashes, indicating that municipalities with better infrastructure enjoyed safer traffic conditions. The municipal human development index (HDI-M) analysis confirmed a relationship between higher socioeconomic development and better road safety conditions. These findings align with the global trend (WHO, 2023), where wealthier countries have lower traffic fatality rates.

SUS emergency units positively correlated with the number of deaths on public roads, suggesting that areas with higher traffic crashes may receive more emergency care units due to their increased need. This relationship highlights the occurrence of traffic crashes and the presence of emergency units could be influenced by underlying factors such as vehicle and infrastructure conditions. While this correlation does not imply causation, it points to the potential interdependence between emergency care infrastructure and crash severity. No significant correlation was found between the number of healthcare facilities and infrastructure investments. Population density was significant, but the models had low parameter values.

Generally speaking, the same behaviors were observed in the variables in the OLS and TOBIT models. Thus, the variables motorcycles and scooters per capita, HDI-M, road extension per municipality, investments in transportation per capita, investments in

housing and urbanism, demographic density, and SUS emergency units were characterized as being the ideal variables to explain the dependent variable in this study. Therefore, these variables were selected for the next study stage, with the application of neural networks.

The regression model validation occurred through the hypothesis tests described for the model conditions. In this way, the collinearity of the variables in the models was tested using the variance inflation factor (VIF). The results showed no values higher than five in the applied set (Table 4), indicating that there was no collinearity among the variables. Also, the normality tests of the residuals showed a normalized trend in the histograms for the Brazil and Northeast OLS models (Figure 3a,b). However, the normality of the model residuals was not supported, as indicated by the high test statistics (2900.995 and 21,912.350) and very low *p*-values for the normality tests, with $\alpha = 5\%$. This result suggested that the residuals did not follow a normal distribution.

For the TOBIT regression models, the normality tests of the residuals had improved values, with chi-squares values of 28.1174 and 34.7124 and *p*-values < 0.001 for both Brazil and the Northeast. In addition, for OLS, the heteroscedasticity of the data was tested with White's test, obtaining *p*-values < 0.001 for the Brazil and Northeast models, with the null hypothesis that the model did not have a dispersion of the errors along the observations presented.

However, since the individual *p*-values for each variable were within the tolerance range for model stability, the responses of their coefficients could not be ruled out, given the coefficients of determination of 0.7437 and 0.6151 for the Brazil and Northeast OLSs.

*3.2. Artificial Neural Network Models*

During the variable selection process, 16 variables were initially considered. The regression analysis (OLS and TOBIT) resulted in the selection of seven significant variables. However, for the artificial neural network (ANN) model, an eighth variable, "cars per capita", was included. Although this variable showed collinearity in the variance inflation factor (VIF) test during regression analysis, it demonstrated solid predictive power and better generalization and prediction performance in the ANN. Therefore, the inclusion of "cars per capita" in the ANN model was justified by its significant contribution to the model's accuracy, leading to using eight variables in the ANN instead of the seven variables used in OLS and TOBIT regression.

We used neural networks and different architectures to train the models in the analyses of Brazil and the Northeast. The network configuration (Table 7) was tested and evaluated for six architectures (A to F), which differed in the number of layers and the number of neurons and activation function per layer, resulting in the best training response and greater accuracy in the test, characterizing appropriate behavior for the data.

**Table 7.** Configurations of the network architectures applied in each model.

| Architecture | 1st Layer Intermediate | | 2nd Layer Intermediate | | 3rd Layer Intermediate | | Use in the Model | |
|:---:|:---:|:---:|:---:|:---:|:---:|:---:|:---:|:---:|
| | No. of Neurons | Activation Function | No. of Neurons | Activation Function | No. of Neurons | Activation Function | Brazil | Northeast |
| A | 10 | Linear | 1 | Linear | - | - | X | X |
| B | 10 | * | 1 | * | - | - | X | X |
| C | 10 | Linear | 1 | * | - | - | X | X |
| D | 10 | Linear | 10 | Linear | 1 | Linear | X | X |
| E | 15 | Linear | 15 | Linear | 1 | Linear | X | - |
| F | 10 | * | 10 | * | 1 | * | - | X |

* Sigmoid hyperbolic tangent.

The architectures were tested using 30% of the unused sample data in the training. R represents the number of successes of the neural network with a margin of $\pm 15\%$ about the actual expected output value, in this case, the number of deaths on public roads in the municipalities. Table 8 presents a summary of the metrics found in each architecture tested.

**Table 8.** Metrics of the ANN architectures applied to analyzing occurrences in the Northeast region and Brazil.

| Metric Brazil | ANN Architectures | | | | |
|---|---|---|---|---|---|
| | A | B | C | D | E |
| $R_T$raining | 0.86217 | 0.5467 | 0.75952 | 0.86244 | 0.86197 |
| F1 Score | 0.41 | 0.37 | 0.59 | 0.88 | 0.87 |
| **Metric Northeast Region** | **ANN Architectures** | | | | |
| | A | B | C | D | F |
| $R_T$raining | 0.7822 | 0.87094 | 0.75381 | 0.78467 | 0.83798 |
| F1 Score | 0.89 | 0.95 | 0.76 | 0.89 | 0.96 |

The best forecast results in Brazil and the Northeast were obtained with the models with a linear activation function and three intermediate layers and the one with two hidden layers. These models corresponded to architectures D and B, where a hit rate of 88% was observed for the Brazil model and 95% for the Northeast model. Figure 4 shows both architectures.

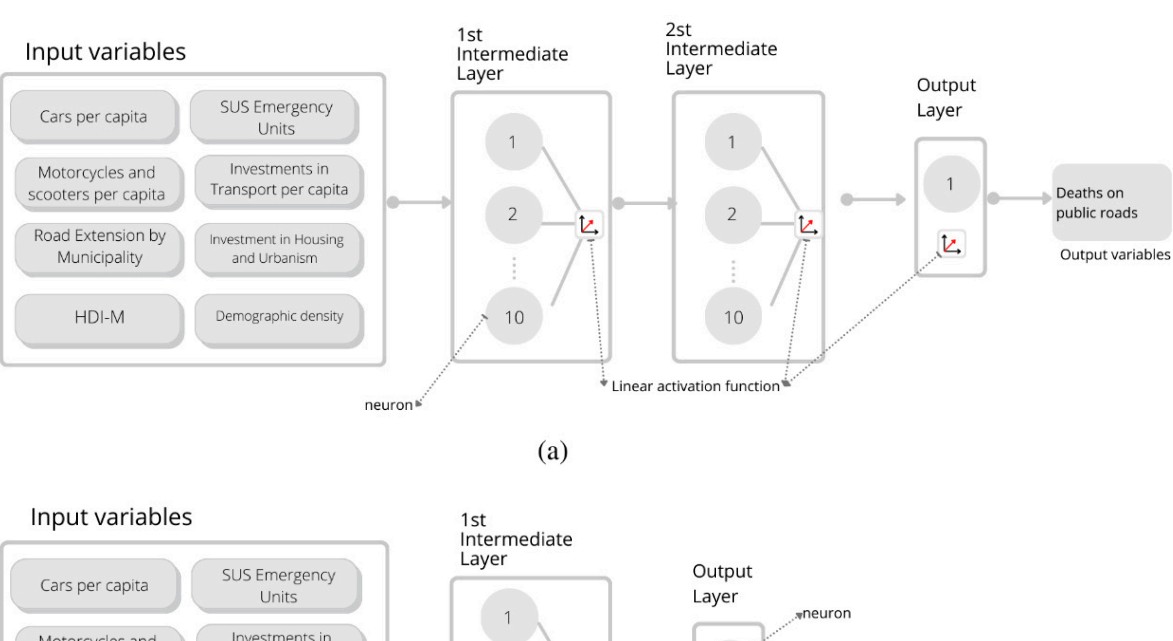

(a)

(b)

**Figure 4.** Best-performing network architectures: (**a**) Brazil analysis model and (**b**) Northeast analysis model.

Despite the F net configuration obtaining a higher percentage of correct predictions for the Northeast, a net with a more complex architecture, be it with a higher number of neurons or layers, demands machine performance that does not justify the higher demand and processing time for the 1% higher efficiency than that of the simpler B model.

As fundamental limitations, the adopted model represents an aggregation of all municipalities in a naturally heterogeneous country, considering for testing only the theoretical variables that influence traffic crashes that are available in each analyzed unit.

## 4. Discussion

Sixteen variables, including socioeconomic and infrastructure, were used to predict the number of deaths on public roads due to traffic crashes. The regression models showed that the most important variables were the number of motorcycles and scooters per capita and the HDI-M, in addition to the number of emergency units, but to a lesser extent.

The motorcycle and scooter per capita variable explained the occurrence of deaths, with higher parameter values for both samples. However, it was higher in the Northeast region in Brazil. This result corroborates the relationship between the higher number of motorcycles in the fleet and high rates of traffic deaths in Brazilian municipalities found by Campos et al. [33]. However, it also reveals the diversity existing in the national context.

The literature reveals that the main factors that contribute to involvement in crashes with motorcycles are human factors, road and vehicle conditions, and environmental factors [54,55], which result in a higher risk of motorcyclists being traffic victims [56–58] since motorcycles are smaller vehicles, which can hinder their visibility in traffic, in addition to not having a vehicle mass that protects the human body. This has also been observed in other countries [59–62].

In this study, the parameters of the regression models indicated the greater importance of the motorcycle fleet for the Northeast region. Motorcycles represent more than 40% of the vehicle fleet in the Northeast [39]. Additionally, it stands out by having the highest traffic morbidity and mortality [63], mainly affecting young men [63,64]. According to de Oliveira et al. [39], the motorcycle is considered an agent of social inclusion, especially for the rural population of the Brazilian northern and northeastern regions, allowing access to services such as health and education. However, given the motorcycle-related exposure and severity observed in different countries, Siman-Tov et al. [65] question the social and economic advantages of two-wheeled vehicles.

The average municipal human development index (HDI-M) had the second-highest positive coefficient in the regression models for the national and regional settings. The results corroborate those of Salehi et al. [66], who verified the relationship between the HDI and traffic crash deaths. The authors found that a high HDI reduces the number of deaths in developed countries. In contrast, this relationship is inverse in developing countries, such as Brazil. In other words, in Brazil, the impact is positive; that is, the number of deaths increases as the HDI increases. The result of the correlation analysis between the HDI-M and traffic fatalities corroborates this finding. The northeastern region had the lowest HDI-M and showed a higher positive correlation with the dependent variable.

The regression coefficients referring to the number of SUS emergency units were significantly positive; the higher the number of units, the higher the number of deaths. Regarding this information, it should first be noted that emergency units are related to the post-crash moment, as presented by the Haddon matrix. As such, other factors—human, vehicle, and roadway—can influence the severity of a crash.

Most traffic crashes occur in regions far from major emergency hospitals, which can lead to longer response times and more deaths [67]. According to Cabral et al. [68], adequate health coverage in more remote areas is important to reduce the likelihood of deaths.

Regarding the results obtained with neural network prediction, the model for the Northeast region had a better response. It is important to emphasize that, despite the F net configuration obtaining a higher percentage of correct answers for the Northeast region, this net had a more complex architecture, with a more significant number of neurons or layers, which required higher computational effort and did not justify its use.

The combined analysis of the three methods, ordinary least squares (OLS) regression, the TOBIT model, and the artificial neural network, highlights the importance of analyzing the relationship between the parameters selected as the input for prediction systems. Most current studies have sought to predict outcomes using various machine learning methods; the combination of these, or the correlation between input variables and delay, has taken place theoretically only, not delving into the data [69].

## 5. Conclusions

This study used neural networks to model the number of deaths occurring at crash sites in municipalities, represented at a macro level. Additionally, this study aimed to understand which macro factors could be related to deaths at crash sites. Focusing on municipal-level data provided broad insights but could have masked micro-level dynamics crucial for targeted interventions.

As limitations, the iForest algorithm excluded outliers, which carries the risk of filtering out extreme but meaningful data points; the Pearson correlation used for descriptive and exploratory analysis to select variables for the regression models may not have fully captured the non-linear relationships within the data; and the OLS and TOBIT regression models analyzed the significance and importance of the independent variables, culminating in predictions with neural networks.

When evaluating factors related to severe crashes resulting in death at the scene, variables such as motorcycles and scooters per capita, HDI-M, and SUS emergency units showed greater explanatory power at the macro level. The proposed methodological approach enabled the neural network to achieve high prediction accuracy rates (above 88%). This approach benefits strategic decision making in areas with limited data reliability or aggregation. However, the diversity observed in Brazil, with a 95% accuracy rate in the Northeast region, hinted at significant regional differences. The model's performance highly depended on quality, consistency, and data management across diverse regions. This implied a limitation regarding regional data heterogeneity, where differences in reporting practices or data aggregation could affect the overall model's reliability.

The artificial neural network (ANN) models outperformed the linear models in predicting fatal traffic crashes due to their ability to capture complex and non-linear patterns. Adjustments to the network architecture, including hidden layers, neurons, and activation functions, combined with cross-validation, resulted in more accurate predictions. Although ANNs are less interpretable, their superior predictive performance is crucial for road safety policies and emergency resource allocation. For decisionmakers and policy formulators, the nature of the ANN's responses may limit the understanding of how specific predictors influence fatality rates, potentially complicating the translation of results into concrete safety measures.

While this study is based on traffic crash data in Brazil, the methodologies and models discussed have potential applicability to other regions with similar demographics and geographic conditions. Future research should explore how these models can be adapted or validated in different contexts, considering local data availability and regional factors. Recommendations for future studies include evaluating the use of weighted emergency units by population or road network and examining the impact of the time between the crash and emergency services' arrival. These limitations offer valuable directions for future

research. They highlight the need for improved data management practices, methods to enhance model interpretability, careful handling of outliers, and more granular and temporally consistent data. Expanding the variable range could further refine predictive accuracy and policy relevance.

**Author Contributions:** Conceptualization, C.F.A.d.S.; methodology, C.F.A.d.S. and H.d.S.Q.J.; software, C.F.A.d.S. and H.d.S.Q.J.; validation, C.F.A.d.S. and M.O.d.A.; formal analysis, C.F.A.d.S.; investigation, C.F.A.d.S.; resources, C.F.A.d.S.; data curation, C.F.A.d.S.; writing—original draft preparation, C.F.A.d.S., M.O.d.A., C.I.d.C., A.M.d.S., H.d.S.Q.J. and V.A.F.; writing—review and editing, C.F.A.d.S., M.O.d.A., C.I.d.C., A.M.d.S., H.d.S.Q.J. and V.A.F.; visualization, C.F.A.d.S.; supervision, C.F.A.d.S.; project administration, C.F.A.d.S.; funding acquisition, C.F.A.d.S. All authors have read and agreed to the published version of the manuscript.

**Funding:** This research received no external funding.

**Data Availability Statement:** The data and materials used in this study will be made available upon request from the corresponding author.

**Conflicts of Interest:** The authors declare no conflicts of interest.

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
