# Peer review of "Macro-Level Modeling of Traffic Crash Fatalities at the Scene: Insights for Road Safety"

_infrastructures, doi:10.3390/infrastructures10050117_

Round 1
Reviewer 1 Report
Comments and Suggestions for Authors
The paper is titled: Macro-Level Predictors of Traffic Crash Fatalities at the Scene: Insights for Road Safety
The methodology defines itself as a prediction. This process usually includes a time dimension. This work is based on data of a specific year; therefore, it should be rather called “modeling” instead of “prediction”.
It is based on a remarkable amount of data, as well as on advanced statistical methods.
This reviewer has a few more concerns, recommended to be addressed in the paper.
The paper uses a lot of explanatory variables. However, some of them are not clear to me. Please explain “Deaths by occurrence”, if possible, in line with WHO classification of death causes.
Also, please define “highway”. Is it a specific road class? Does it relate only to inter-urban sections, or are urban roads included?
Please check the content of “investment”. Is it really (capital) investment or is it rather expenditure on transport/housing?
Author Response
Dear Reviewer,
We sincerely thank you for your careful reading and thoughtful feedback on our manuscript. We greatly appreciate your recognition that the introduction, results, and conclusion are adequately described. We also recognize that the research design and methods need to be improved. Thus, we focus on clarifying and answering the questions raised and improving the quality of our work. We also appreciate your statement that the English text is fine and does not require any improvement.
- The paper is titled: Macro-Level Predictors of Traffic Crash Fatalities at the Scene: Insights for Road Safety. The methodology defines itself as a prediction. This process usually includes a time dimension. Since this work is based on data from a specific year, it should be called “modeling” instead of “prediction.”
We appreciate your observation and understand the importance of aligning the title with the objectives and research design. Accordingly, we revised the title to “MACRO-LEVEL MODELING OF TRAFFIC CRASH FATALITIES AT THE SCENE: INSIGHTS FOR ROAD SAFETY” and changed the idea of prediction to modeling in many parts of the manuscript to better reflect the central theme developed in the manuscript. Examples of these shifts are as follows:
Abstract: This paper applies 2019 macro-level data from DATASUS to model traffic fatalities at the scene. The methodology used Ordinary Least Squares (OLS) and censored regression models (TOBIT) to identify the significant variables explaining the occurrence of deaths on public roads due to crashes. The number of fatalities on public roadways was then modeled using a multilayer perceptron Artificial Neural Network employing the significant variables as predictors according to the generalization capacity of complex predictive models. The OLS and TOBIT findings have indicated that the variables Motorcycles and Scooters per capita, Municipal Human Development Index, and Number of SUS Emergency Units are the most important for modeling traffic fatalities at the scene at the national and regional levels. Applying these variables, the neural network's best results achieved a hit rate of 88% for Brazil and 95% for the Northeast model. The contribution lies in an approach combining various methods and considering a range of variables influencing traffic fatalities at the scene. The findings offer insights for policymakers, researchers, and practitioners involved in road safety initiatives, mainly where crash data is scarce and macro-level analysis is necessary. (lines 13-32)
The response variable used in this study corresponds to the number of road deaths due to traffic crashes in Brazilian municipalities in 2019. These data were obtained from the Mortality Information System of the Unified Health System (DATASUS) and originated from the "Death Declaration" document. The indices of traffic deaths per inhabitant and vehicle allow us to know the negative impact of traffic crashes and measure traffic safety [44]. In this way, modeling the number of deaths makes it possible to identify the macro-level factors related to more severe crashes, identifying the socioeconomic and road infrastructure characteristics of the most exposed municipalities. (lines 160-168).
SUS Emergency Units positively correlated with the number of deaths on public roads, suggesting that areas with higher traffic crashes may receive more emergency care units due to their increased need. This relationship highlights the occurrence of traffic crashes, and the presence of emergency units could be influenced by underlying factors such as vehicle and infrastructure conditions. While this correlation does not imply causation, it points to the potential interdependence between emergency care infrastructure and crash severity. No significant correlation was found between the number of healthcare facilities and infrastructure investments. Population density is significant, but the models reveal low parameter values. (lines 362 -370)
This paper used neural networks to model the number of deaths occurring at crash sites in municipalities, represented at a macro level. Additionally, the study aimed to understand which macro factors could be related to deaths at the crash site. Focusing on municipal-level data provides broad insights but may mask micro-level dynamics crucial for targeted interventions. (lines 505 – 509)
- It is based on a remarkable amount of data, as well as on advanced statistical methods. This reviewer has a few more concerns, recommended to be addressed in the paper. The paper uses a lot of explanatory variables. However, some of them are not clear to me. Please explain “Deaths by occurrence”, if possible, in line with WHO classification of death causes.
“Deaths by occurrence rate” – Deaths by occurrence divided by the popula-tion of a given area, then multiplied by 100,000 inhabitants. This variable refers to the number of deaths resulting from transport-related accidents, divided by the population of a given area, and multiplied by 100,000 inhabitants. It follows the classification of the International Classification of Diseases (ICD-10), Chapter XX, codes V01–V99, as adopted by the Brazilian Ministry of Health. The indicator enables standardized comparison of mortality rates due to crash accidents across different geographic areas.
- Also, please define “highway”. Is it a specific road class? Does it relate only to inter-urban sections, or are urban roads included?
There is indeed some confusion between "road" and "highway" in the text and in the variable titles. We have reviewed the text and tables and standardized everything to the word "road," which better represents the research, as it refers to the road network rather than the intra-urban network or urban streets. Examples of these shifts are as follows:
3.1 Brazil has an estimated 2020 population of 212.6 million inhabitants [34] in an area of 8,510,345.538 km² (Figure 1). The country's federal road network accounted for a total length of 75.8 thousand km in 2019, of which 65.4 thousand km corresponded to paved roads [35]. Regarding its vehicular fleet, Brazil registered a total of 107,948,371 vehicles (cars, light commercials, trucks, buses, motorcycles, scooters, and others) in 2020, according to the National Traffic Secretariat [36]. (lines 127 – 132).
3.2 Changes in Table 2.
|
Road Extension by cars |
Total road mileage divided by the number of cars |
2019 |
km/car |
[42,36] |
|
Road Extension by Motorcycles |
Total road mileage divided by the number of motorcycles |
2019 |
km/ motorcycle |
[42,36] |
|
Road Extension by Municipality |
Length of roads per municipality Road Mileage in km |
2019 |
km |
[42,40] |
|
Road Density |
Road Mileage in km per inhabitant |
2020 |
km/inhabitant |
[42,40] |
|
Investment in Road Infrastructure per capita |
Monetary values (in Brazilian Real) invested in road infrastructure per inhabitant |
2020 |
R$/inhabitant |
[43] |
Please check the content of “investment”. Is it really (capital) investment or is it rather expenditure on transport/housing?
Investment in this article includes capital investments in transportation infrastructure as well as expenditures in maintenance and operational costs of the assets.

Reviewer 2 Report
Comments and Suggestions for Authors
I rate the originality of the study as average for the following reasons:
a) Models of accident prediction on a macro scale are a well-recognized problem. The explanatory variables appearing in these models are usually repeated in different studies.
b) the authors' exemplary statement that the significant explanatory variables in the model of predicting the number of fatalities are: motorcycles and scooters per capita, HDI-M, Road Extension per Municipality, investments in transportation per capita, investments in housing and urbanism, demographic density, does not contribute many new elements of knowledge. The only surprising thing is the described impact of SUS emergency units. However, the authors explain this surprising result of the analyses.
c) the approach of combining various methods and considering a range of variables influencing traffic fatalities at the scene is an interesting approach, but it does not go beyond the so-called good practice in scientific research.
The assignment of the “average” rating in Significance of Content is partly related to the comments given above. The results of the analyses themselves are to some extent obvious, because they link the number of fatalities to various measures of risk exposure. The study shows the regional specificity of the impact of demographic and economic conditions on road safety. However, this may limit the interest to a group of readers from the South and Central American region. On the other hand, I consider the good presentation of the analysis methods in adaptation to the collected data on accident victims and characteristics of municipalities and regions, as well as data on motorization, to be valuable and of greater importance.
The structure of the study and the details of the description are beyond reproach. They are legible and well-present the most important elements of the described research. The discussion of the results of the analyses and the conclusions are correct.
The assignment of the "average" rating in the scope of Scientific Soundness results from the good structure of the study and good documentation of the analysis results. The authors' diligence in conducting the research should be appreciated, but the analysis methods themselves can be described as classic and do not bring new ideas.
Author Response
Reviewer #2
We sincerely thank you for your careful reading and thoughtful feedback on our manuscript. We greatly appreciate your recognition that the introduction, research design, methodologies, results, and conclusion are adequately described. We also appreciate your statement that the English text is fine and does not require any improvement.
- I rate the originality of the study as average for the following reasons:
a) Models of accident prediction on a macro scale are a well-recognized problem. The explanatory variables appearing in these models are usually repeated in different studies.
We appreciate your assessment and acknowledge that accident prediction models on a macro scale are indeed a well-recognized area of research. However, while similar explanatory variables are often used across different studies, our work brings a unique perspective by applying these methodologies in the context of Brazil and South America, regions where macro-scale accident prediction remains largely underexplored. By focusing on this geographic area, we aim to contribute valuable insights and expand the applicability of such models to new contexts.
- The authors' exemplary statement that the significant explanatory variables in the model of predicting the number of fatalities are: motorcycles and scooters per capita, HDI-M, Road Extension per Municipality, investments in transportation per capita, investments in housing and urbanism, demographic density, does not contribute many new elements of knowledge. The only surprising thing is the described impact of SUS emergency units. However, the authors explain this surprising result of the analyses.
We appreciate the reviewer’s observation. Our study confirms that most of the significant explanatory variables—such as motorcycles and scooters per capita, HDI-M, road extension per municipality, investments in transportation per capita, investments in housing and urbanism, and demographic density—follow an expected logic based on the established literature on accident prediction models. These findings reinforce that our macro-scale approach, applied to Brazil and South America, produces results consistent with what has been observed in similar contexts. The only surprising result was the impact of SUS emergency units. We acknowledge this unexpected outcome and have dedicated a section of our analysis to thoroughly explain it. We suggest that the counterintuitive effect might be related to emergency units related to the municipalities' population, since they also showed a high and positive correlation with the variable’s road density and demographic density. This explanation underscores that while most of our findings align with expectations, the nuanced interpretation of the SUS emergency unit’s variable enriches the discussion and contributes valuable insights to the literature.
The emergency units, on the other hand, may be related to the municipalities' population since they also showed a high and positive correlation with the variable’s road density (11) and demographic density. (lines 323 – 325).
- The approach of combining various methods and considering a range of variables influencing traffic fatalities at the scene is an interesting approach, but it does not go beyond the so-called good practice in scientific research.
The assignment of the “average” rating in Significance of Content is partly related to the comments given above. The results of the analyses themselves are to some extent obvious, because they link the number of fatalities to various measures of risk exposure. The study shows the regional specificity of the impact of demographic and economic conditions on road safety. However, this may limit the interest to a group of readers from the South and Central American region. On the other hand, I consider the good presentation of the analysis methods in adaptation to the collected data on accident victims and characteristics of municipalities and regions, as well as data on motorization, to be valuable and of greater importance.
We appreciate the reviewer’s recognition of our comprehensive approach, which combines multiple methods and various explanatory variables. Our aim was not necessarily to reinvent methodological approaches but to apply well-established techniques to an underexplored context. While the association between risk exposure and the number of fatalities may seem expected, validating these relationships within the macro-scale framework of Brazilian and South American regions provides new context-specific insights that are essential for regional policy and academic debate.
Furthermore, although some results may appear obvious at first glance, they confirm the robustness of our analytical framework. By demonstrating the regional specificity of the impact of demographic and economic conditions on road safety, our study reinforces the importance of context in interpreting risk factors. This nuance is often overlooked in more generic analyses. This detailed presentation of our methods and data adaptation, particularly regarding accident victim profiles and regional characteristics, adds significant value to the literature, especially for readers interested in the particular dynamics of South and Central America.
Lastly, our analysis thoroughly explains the unexpected findings, such as the impact of the emergency units in the public health system, though singular. This further illustrates our commitment to understanding both anticipated and surprising results, ultimately enhancing our contributions to knowledge in the field of traffic safety.
- The structure of the study and the details of the description are beyond reproach. They are legible and well-present the most important elements of the described research. The discussion of the results of the analyses and the conclusions are correct.
Thank you very much for your kind remarks. We truly appreciate your positive feedback on the structure and clarity of our study. It is gratifying to learn that you found our descriptions precise and that the presentation of the core elements of our research is both legible and comprehensive. We are also pleased that you consider the discussion of the analysis results and conclusions correct. Your encouragement is invaluable and motivates us to continue producing high-quality research.
- The assignment of the "average" rating in the scope of scientific soundness results from the study's good structure and good documentation of the analysis results. The authors' diligence in conducting the research should be appreciated, but the analysis methods themselves can be described as classic and do not bring new ideas.
We appreciate the reviewer's acknowledgment of our study’s robust structure and the thorough documentation of our analysis results. While we understand that the methods employed are considered classic and do not introduce groundbreaking techniques in themselves, we believe that applying these established methods in a macro-scale context—especially within Brazil and South America—offers substantial scientific value. This regional focus not only reinforces known relationships between risk exposure and fatality rates but also highlights nuanced impacts, such as the surprising influence of SUS emergency units, which we have carefully explained in our analysis.
In synthesizing the previous comments, we recognize that while some aspects of our work may appear expected, the diligence and methodological rigor underpinning the study contribute significantly to the literature. The clear presentation of our analysis methods, the detailed adaptation of data on accident victims and municipal characteristics, and the illumination of the regional specificity of demographic and economic conditions on road safety are all significant contributions. These facets underscore that, despite relying on classic methods, our research fills a gap in the macro-scale analysis of traffic fatalities in a largely underexplored geographic context.
We are grateful for the balanced feedback, which reinforces that our study not only adheres to good scientific practice but also provides critical insights that can inform both policy and further research in the field of road safety.
Best regards
